# PDQ-Net: Deep Probabilistic Dual Quaternion Network for Absolute Pose Regression on $SE(3)$

**Wenjie Li**[1]     **Wasif Naeem**[2]     **Jia Liu**[1]     **Dequan Zheng**[1]     **Wei Hao**[1]     **Lijun Chen**[1]

[1]Department of Computer Science and Technology, Nanjing University, Nanjing, China
[2]School of Electronics, Electrical Engineering and Computer Science, Queen's University Belfast, Belfast, UK

## Abstract

Accurate absolute pose regression is one of the key challenges in robotics and computer vision. Existing direct regression methods suffer from two limitations. First, some noisy scenarios such as poor illumination conditions are likely to result in the uncertainty of pose estimation. Second, the output n-dimensional feature vector in the Euclidean space $\mathbb{R}^n$ cannot be well mapped to $SE(3)$ manifold. In this work, we propose a deep dual quaternion network that performs the absolute pose regression on $SE(3)$. We first develop an antipodally symmetric probability distribution over the unit dual quaternion on $SE(3)$ to model uncertainties and then propose an intermediary differential representation space to replace the final output pose, which avoids the mapping problem from $\mathbb{R}^n$ to $SE(3)$. In addition, we introduce a backpropagation method that considers the continuousness and differentiability of the proposed intermediary space. Extensive experiments on the camera re-localization task on the Cambridge Landmarks and 7-Scenes datasets demonstrate that our method greatly improves the accuracy of the pose as well as the robustness in dealing with uncertainty and ambiguity, compared to the state-of-the-art.

## 1 INTRODUCTION

The absolute pose estimation refers to inferring the object's pose (i.e., position and orientation) in 3D space from 2D input images, which is an age-old problem in the fields of robotics [Mur-Artal and Tardós, 2017, Wang et al., 2018, Campos et al., 2021] and computer vision [Kendall et al., 2015, Deng et al., 2019, Turkoglu et al., 2021]. Early research focuses on geometry-based approaches due to their reliability and accuracy in some static environments. How-ever, the geometry-based approaches cannot function well in some specific scenarios such as poor illumination and the textureless case.

In recent years, the deep learning technique has provided us with an alternative vehicle to regress the absolute pose. In these approaches, a proper output for representing the pose is particularly important when dealing with more complicated environments. A number of deep models have been presented to output an n-dimensional feature vector that is used to directly denote the pose, where the rotation is mostly represented by the Euler angle, unit quaternion, or rotation matrix, and translation is substituted by a 3-dimensional vector [Wang et al., 2018, Sattler et al., 2019].

In spite of some advancements, these learning-based techniques suffer from one or two limitations. First, some work cannot capture and model pose uncertainties in practical scenarios, which affects the estimation accuracy. Second, an n-dimensional feature vector on a Euclidean space $\mathbb{R}^n$ is taken as the common output by existing work [Shotton et al., 2013, Sattler et al., 2019, Xue et al., 2019], which however cannot be well mapped to $SE(3)$ manifold since the pose in $SE(3)$ is not homeomorphic to the Euclidean space $\mathbb{R}^n$ [LaValle, 2006]. For example, the angle 0 and $2\pi$ in $\mathbb{R}^n$ can map to the same rotation in $SE(3)$, which fails to satisfy the one-to-one mapping feature in homeomorphism theory.

In this paper, we propose a probabilistic deep dual quaternion network that regresses the absolute pose on $SE(3)$ from a single RGB image. Compared to existing learning-based approaches, our model has three advanced features. First, we take pose uncertainties into consideration by introducing an antipodally symmetric distribution over the unit dual quaternion on $SE(3)$. In this way, the pose regression problem can be converted into a deep probabilistic problem. Second, we propose to estimate the pose indirectly by presenting an intermediary differential representation space as the output of our deep probabilistic model. Afterward, the rotation and translation can be derived from the inter-

*Accepted for the 38[th] Conference on Uncertainty in Artificial Intelligence* (UAI 2022).

mediary representation space by modeling a quadratically-constrained quadratic program (QCQP) problem and a Gaussian process respectively, which avoids the issue of mapping from $\mathbb{R}^n$ to $SE(3)$. Third, we introduce a backpropagation method that considers the continuousness and differentiability of the proposed intermediary space. Extensive experiments on the camera re-localization task on the Cambridge Landmarks and 7-Scenes datasets demonstrate that our method outperforms the state-of-the-art, in terms of pose accuracy and robustness.

In summary, our work makes the following contributions:

- We develop an antipodally symmetric probability distribution over the unit dual quaternion on $SE(3)$ to model the pose uncertainty, which is the key factor in improving the accuracy of pose regression.

- We propose a deep dual quaternion model to regress the pose indirectly, which effectively addresses the mapping problem from $\mathbb{R}^n$ to $SE(3)$. Additionally, a backpropagation method is introduced to the proposed model.

- We implement our deep probabilistic model on the Cambridge Landmarks and 7-Scenes datasets. Extensive experiment results show that our method greatly improves the accuracy of the pose as well as the ability in dealing with uncertainty and ambiguity, compared to the state-of-the-art.

## 2 RELATED WORK

Absolute pose estimation combining with the deep learning technique becomes a hot topic in robotics and computer vision field. The related techniques roughly fall into two categories: direct pose regression and indirect pose regression.

Direct pose regression aims to directly regress the absolute pose from sequential RGB images by utilizing various well-designed end-to-end networks [Wang et al., 2018, Clark et al., 2017, Kendall et al., 2015, Shotton et al., 2013, Chen et al., 2021]. In these approaches, most of them follow the same pipeline: features are extracted using a defined network such as the PoseNet [Kendall et al., 2015], the Branch-Net [Wu et al., 2017], which are then embedded into a high-dimensional vector that lies in the Euclidean space $\mathbb{R}^n$. Then Sattler et al. [2019] pointed out that this embedding layer typically corresponds to the output of the second-to-last layer in direct pose regression methods. The last layer performs a linear projection from the embedding space to the space of poses. However, these methods commonly suffer from one of two issues. First, the uncertainty of poses may result in the degradation of the accuracy of predicted poses since some of them are not robust enough to capture uncertainties [Kendall and Cipolla, 2016]. Second, the output n-dimensional feature vector lying on a Euclidean space $\mathbb{R}^n$ may not be well mapped to $SE(3)$ [LaValle, 2006].

A strategy to overcome these problems is to regress the absolute pose indirectly, which is achieved by regressing an intermediary vector to replace the output feature vector [Deng et al., 2020, Bui et al., 2020]. However, it is usually hard to find such a representation space. Poursaeed et al. [2018] introduced a Siamese model for uncalibrated cameras to regress a fundamental matrix that serves as an intermediary representation of camera poses, but it fails to capture uncertainties of poses. Recently, deep probabilistic models have been developed by regressing essential parameters of the probabilistic distribution. Gilitschenski et al. [2020] introduced a deep Bingham model for the object orientation estimation on $SO(3)$ by regressing the orthogonal matrix of the Bingham distribution, where pose uncertainties are modeled as a Bingham distribution. Mohlin et al. [2020] similarly developed a deep matrix Fisher distribution for object rotation estimation on $SO(3)$ by regressing the parameter matrix of the Fisher distribution. But it is generally hard to find such statistic distributions on $SE(3)$ manifold to measure pose uncertainties.

To this point, we develop an antipodally symmetric probability distribution over the unit dual quaternion on $SE(3)$ to model pose uncertainties. Based thereon, we present a deep probabilistic distribution to indirectly regress the absolute pose on $SE(3)$.

## 3 UNIT DUAL QUATERNION DISTRIBUTION

This section gives the definition of the unit dual quaternion distribution on $SE(3)$. For this purpose, we first briefly revisit the concept of dual quaternions and then give the description of the unit dual quaternion distribution.

### 3.1 DUAL QUATERNION

In this work, a quaternion $\mathbf{q}$ is defined as $\mathbf{q} = q_0 + q_1\mathbf{i} + q_2\mathbf{j} + q_3\mathbf{k}$, the $\{\mathbf{i}, \mathbf{j}, \mathbf{k}\}$ is the standard basis of the three-dimensional Euclidean space $\mathbb{R}^3$. For convenience, we bring the vector $\mathbf{q} = [q_0, \mathbf{q}_{vec}] \in \mathbb{R}^4$ to denote the quaternion. The multiplication between two arbitrary quaternions can be done with a matrix-vector form that is given by

$$\mathbf{p} \odot \mathbf{q} = \mathbf{R}_q\mathbf{p} = \begin{bmatrix} q_0 & -\mathbf{q}_{vec}^T \\ \mathbf{q}_{vec} & -\mathbf{q}_{vec}^{\times} + q_0\mathbf{I}_3 \end{bmatrix} \mathbf{p},$$

where $[\mathbf{a}]^{\times}$ denotes the skew-symmetric matrix formed from the vector $\mathbf{a}$, and $\mathbf{I}$ refers to the identity matrix.

The norm of a quaternion is defined as $\sqrt{\mathbf{q} \odot \mathbf{q}^*}$, with $\mathbf{q}^* = [q_0, -\mathbf{q}_{vec}]$ being the conjugate of $\mathbf{q}$. And quaternions with a unit norm are called unit quaternions, which are used for denoting the pure rotation on the unit hypersphere $\mathbb{S}^3 \subset \mathbb{R}^4$.

Dual quaternion consists of the real part quaternion and dual part quaternion which is a convenient tool for encapsulating

the rotation and translation [Leclercq et al., 2013],

$$\mathbf{v} = \mathbf{q}_r + \epsilon \mathbf{q}_d, \epsilon \neq 0, \epsilon^2 = 0, \quad (1)$$

where $\mathbf{q}_r$ is the unit quaternion for indicating the rotation, and $\mathbf{q}_d$ is the dual part quaternion for representing the composition of the rotation quaternion and translation quaternion $\mathbf{q}_t = [0, t_x, t_y, t_z]^T$, with $\mathbf{q}_d = 0.5\mathbf{q}_t \odot \mathbf{q}_r \in \mathbb{R}^4$.

Since the dual part $\mathbf{q}_d$ is orthogonal to the real part $\mathbf{q}_r$ on the hypersphere space $\mathbb{S}^3$, we further get the unit dual quaternion manifold $\mathbb{DH}_1 := \{[\mathbf{q}_r^T, \mathbf{q}_d^T]^T \|\|\mathbf{q}_r\| = 1, \mathbf{q}_r \in \mathbb{S}^3, \mathbf{q}_r^T \mathbf{q}_d = 0\} \subset \mathbb{R}^8$. Furthermore, the translation vector $\mathbf{t} = [t_x, t_y, t_z]$ can be recovered from $\mathbb{DH}_1$ according to [Li et al., 2021], which can be written as

$$\mathbf{t} = 2[\mathbf{R}_{qr}]_{1:3}^T \mathbf{q}_d, \quad (2)$$

where $[\mathbf{R}_{qr}]_{1:3}$ refers to the last three columns of the right multiplication matrix $\mathbf{R}_{qr}$.

### 3.2 EXPONENTIAL DISTRIBUTION OF UNIT DUAL QUATERNION

Unit dual quaternions $\mathbf{v}$ and $-\mathbf{v}$ denote the same transformation because of the property of unit quaternions, namely $\mathbf{q}_r = -\mathbf{q}_r$. We assume there exists an antipodally symmetric exponential distribution of unit dual quaternions $\mathbf{v} \in \mathbb{DH}_1$ [1]. Previous work [Gilitschenski et al., 2014] offered an exponential distribution for representing the orientation and position on $SE(2)$ manifold but failed to be applied on $SE(3)$. In this work, we bridge this gap by developing the unit dual quaternion distribution on $SE(3)$.

**Definition 1.** *A vector $\mathbf{v} \in \mathbb{DH}_1 \subset \mathbb{R}^8$ can be modeled as an antipodally symmetric distribution if its probability density function has the following form*

$$f(\mathbf{v}) = \frac{1}{N(\mathbf{F})} \exp(\mathbf{v}^T \mathbf{F} \mathbf{v}), \quad (3)$$

*where $N(\mathbf{F})$ refers to the normalization constant of the proposed distribution. And the real symmetric matrix $\mathbf{F} \in \mathbb{R}^{8 \times 8}$ is the parameter matrix.*

We split the vector $\mathbf{v} = [\mathbf{q}_r^T, \mathbf{q}_d^T]^T$ with $\mathbf{q}_r \in \mathbb{S}^3$ and $\mathbf{q}_d \in \mathbb{R}^4$. Meanwhile, we decompose the symmetric parameter matrix $\mathbf{F}$ as follows,

$$\mathbf{F} = \begin{bmatrix} \mathbf{F}_1 & \mathbf{F}_2 \\ \mathbf{F}_2^T & \mathbf{F}_3 \end{bmatrix}, \mathbf{F}_i \in \mathbb{R}^{4 \times 4}, i = 1, 2, 3.$$

---

[1] Generally, the distribution $f$ on the space $S$ is said to be antipodally symmetric if $f(-\mathbf{v}) = f(\mathbf{v})$ for all $\mathbf{v} \in S$, which means the opposite points on $S$ have equal probability.

Then the exponential distribution (3) can be rewritten as

$$f(\mathbf{v}) = \frac{1}{N(\mathbf{F})} \exp \underbrace{(\mathbf{q}_r^T (\mathbf{F}_1 - \mathbf{F}_2 \mathbf{F}_3^{-1} \mathbf{F}_2^T) \mathbf{q}_r}_{Bingham-like} + \underbrace{(\mathbf{q}_d + \mathbf{F}_3^{-1} \mathbf{F}_2^T \mathbf{q}_r)^T \mathbf{F}_3 (\mathbf{q}_d + \mathbf{F}_3^{-1} \mathbf{F}_2^T \mathbf{q}_r))}_{Gaussian-like} . \quad (4)$$

In this distribution, we have the following theorem for parameter matrix $\mathbf{F}$, where the proof can be found in the Supplementary Material.

**Theorem 1.** *Considering the antipodally symmetric distribution (4), the sub-block matrix $\mathbf{F}_1 \in \mathbb{R}^{4 \times 4}$ is real symmetric, and $\mathbf{F}_3 \in \mathbb{R}^{4 \times 4}$ is real symmetric and negative definite.*

## 4 SYMMETRIC MATRIX F ON SE(3)

This section gives a thorough analysis of the parameter matrix $\mathbf{F}$ in the unit dual quaternion distribution on $SE(3)$.

### 4.1 SYMMETRIC MATRIX F

As shown in Equation (4), $\mathbf{F}$ is decomposed into three submatrices, which are significant components of the Binghamlike distribution and the Gaussian-like distribution.

#### 4.1.1 The Bingham-like Matrix

The Bingham distribution was introduced [Bingham, 1974] as an extension of the Gaussian distribution, which lies on the surface of the unit hypersphere,

$$f(\mathbf{q}; \mathbf{M}, \mathbf{Z}) = \frac{1}{N(\mathbf{Z})} \exp \left( \mathbf{q}^T \mathbf{M} \mathbf{Z} \mathbf{M}^T \mathbf{q} \right), \quad (5)$$

where $\mathbf{Z} \in \mathbb{R}^{4 \times 4}$ is a diagonal matrix with an ascending entries $z_1 \leq z_2 \leq z_3 \leq z_4 \leq 0$, the matrix $\mathbf{M} \in \mathbb{R}^{4 \times 4}$ is an orthogonal matrix, and $N(\mathbf{Z})$ is the normalization constant. Usually, we enforce the last entry of $\mathbf{Z}$ as a zero value using the property of the Bingham distribution, namely $f(\mathbf{q}_r; \mathbf{M}, \mathbf{Z}) = f(\mathbf{q}_r; \mathbf{M}, \mathbf{Z} + c\mathbf{I})$, by setting $c = -z_4$.

Here we show that the sub-vector $\mathbf{q}_r \in \mathbb{S}^3$ follows the Bingham distribution, the essential matrix $\mathbf{M} \in \mathbb{R}^{4 \times 4}$ and $\mathbf{Z} \in \mathbb{R}^{4 \times 4}$ can be computed according to the Theorem 2, the proof can be found in the Supplementary Material.

**Theorem 2.** *The parameter matrix $\mathbf{F} \in \mathbb{R}^{8 \times 8}$ is able to be decomposed into an orthogonal matrix $\mathbf{M} \in \mathbb{R}^{4 \times 4}$ and a diagonal matrix $\mathbf{Z} \in \mathbb{R}^{4 \times 4}$ via the eigendecomposition of $\mathbf{F}_1 - \mathbf{F}_2 \mathbf{F}_3^{-1} \mathbf{F}_2^T$.*

$$\mathbf{B} = \mathbf{F}_1 - \mathbf{F}_2 \mathbf{F}_3^{-1} \mathbf{F}_2^T$$

$$\min_{\mathbf{q}_r \in \mathbb{S}^3} \mathbf{q}_r^T \mathbf{B} \mathbf{q}_r$$
$$s.t. \quad \mathbf{q}_r^T \mathbf{q}_r = 1$$
$\pm \mathbf{q}_r$

Figure 1: The differentiable QCQP for representing the rotation.

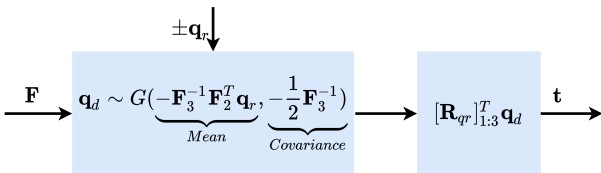

Figure 2: The Gaussian process for representing the translation.

### 4.1.2 The Gaussian-like Matrix

Similarly, the parameter matrix $\mathbf{F}$ is also critical in the marginal distribution of $\mathbf{q}_d$ which is the Gaussian distribution with the mean vector being $-\mathbf{F}_3^{-1}\mathbf{F}_2^T\mathbf{q}_r$ and covariance matrix being $-\frac{1}{2}\mathbf{F}_3^{-1}$,

$$f(\mathbf{q}_d|\mathbf{q}_r) \propto$$
$$\exp\left(\mathbf{q}_d - (-\mathbf{F}_3^{-1}\mathbf{F}_2^T\mathbf{q}_r)\right)^T \mathbf{F}_3(\mathbf{q}_d - (-\mathbf{F}_3^{-1}\mathbf{F}_2^T\mathbf{q}_r)). \quad (6)$$

## 4.2 F AS THE INTERMEDIARY REPRESENTATION SPACE

As $\mathbf{F}$ is an essential element of the developed unit dual quaternion distribution, in this subsection, we show that it can be considered as the intermediary representation space for regressing the pose.

### 4.2.1 Rotation Representation

For convenience, we set the upper case $\mathbf{B} = \mathbf{F}_1 - \mathbf{F}_2\mathbf{F}_3^{-1}\mathbf{F}_2^T$, which is taken as the representation for rotation quaternions. According to the properties of matrix theory, the matrix $\mathbf{B} \in \mathbb{R}^{4\times4}$ is real symmetric with a simple minimum eigenvalue, which is written as

$$\mathbf{B} = \begin{bmatrix} b_1 & b_2 & b_3 & b_4 \\ b_2 & b_5 & b_6 & b_7 \\ b_3 & b_6 & b_8 & b_9 \\ b_4 & b_7 & b_9 & b_{10} \end{bmatrix}.$$

Then we regard the computation of the rotation quaternion $\mathbf{q}_r$ as an optimization problem which is defined as a quadratically-constrained quadratic program (QCQP) problem that arises in [Yang and Carlone, 2019].

**Definition 2** (QCQP problem). *Let matrix* $\mathbf{B} \in \mathbb{R}^{4\times4}$ *be a symmetric matrix, which can be parameterized with the vector* $\mathbf{b} \in \mathbb{R}^{10}$. *The QCQP problem related to* $\mathbf{B}$ *is shown in Figure 1, which is written as*

$$\min_{\mathbf{q}_r \in \mathbb{S}^3} \mathbf{q}_r^T \mathbf{B} \mathbf{q}_r$$
$$s.t. \quad \mathbf{q}_r^T \mathbf{q}_r = 1. \quad (7)$$

Note that the solution to this problem in Figure 1 is to calculate the eigenvector corresponding to the minimum eigenvalue of $\mathbf{B}$.

### 4.2.2 Translation Representation

The parameter matrix $\mathbf{F}$ also offers a new standpoint for representing the translation vector. As known, the marginal distribution of $\mathbf{q}_d$ is a Gaussian distribution which is shown in Equation (6). Hence, the $\mathbf{q}_d$ can be represented by $\mathbf{m} = -\mathbf{F}_3^{-1}\mathbf{F}_2^T\mathbf{q}_r$, with the uncertainty being measured by the covariance matrix $\mathbf{G} = -\frac{1}{2}\mathbf{F}_3^{-1}$, the related Gaussian process is illustrated in Figure 2. Finally the translation can be recovered from $\mathbf{q}_d$ using Equation (2).

## 5 DEEP LEARNING AND F

In this section, we intend to develop a new probabilistic dual quaternion network to indirectly regress the pose. First, we show $\mathbf{F}$ is a smooth representation of the pose. Next, we develop a backpropagation method using an implicit function theorem. Subsequently, a new uncertainty metric is proposed to measure uncertainties. Finally, we give the structure of the proposed network.

### 5.1 SMOOTH FEATURE OF F

A smooth representation for $SE(3)$ is important for learning-based methods when concerning the backpropagation procedure. Here we mainly consider the smooth feature of $\mathbf{F}$ on $SO(3)$ manifold since the translation can be computed after rotation quaternions are estimated.

We utilize the concept of *continuous representation* presented in [Zhou et al., 2019] to give a specific analysis to it. Considering the surjective map between the representation space and original space which is shown in Figure 4, we set the matrix $\mathbf{B}$ as the representation space and the rotation quaternions $\mathbf{q}_r$ as the original space. Zhou et al. [2019] demonstrated that the mapping function $(f, h)$ is a representation if $f$ is a left inverse of $h$. Conversely, the representation is continuous if $h$ is continuous. From the solution of QCQP problem, we show that the original space $\mathbf{q}_r$ can be computed via the eigendecomposition of $\mathbf{B}$, namely

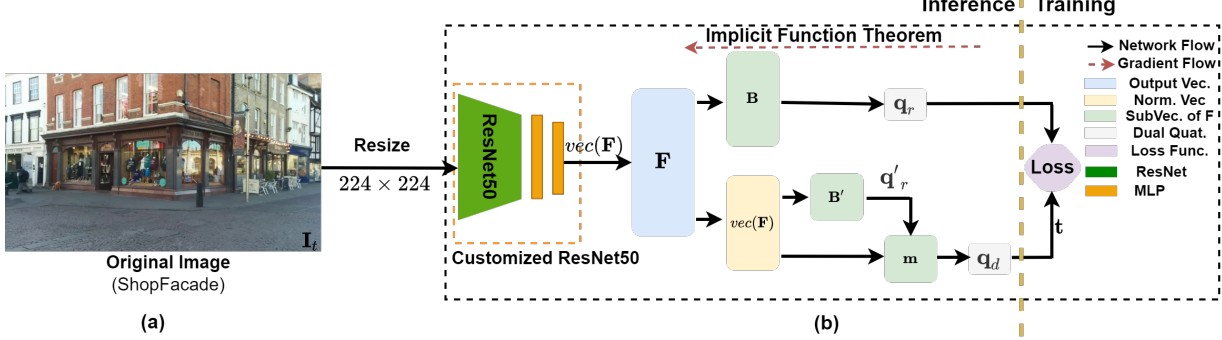

Figure 3: The proposed deep dual quaternion network structure. The letters (a), (b) correspond to the input image, and the proposed deep dual quaternion network. We start from the current frame input $I_t$ which is further fed into the proposed network. The output of the network is a vector $vec(\mathbf{F})$ with 36 elements which then consists of the matrix $\mathbf{F}$. After learning the parameter matrix $\mathbf{F}$, the rotation quaternions $\mathbf{q}_r$ and translation vector $\mathbf{t}$ are computed using the theory of dual quaternion distribution. During training, an implicit function theorem is applied to the backpropagation procedure.

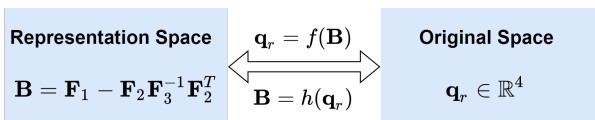

Figure 4: A surjective map between the representation space and the original space.

$\mathbf{q}_r = f(\mathbf{B})$. And the representation space can be reversely deduced from the original space $\mathbf{q}_r$, namely $\mathbf{B} = h(\mathbf{q}_r)$. Moreover, a continuous representation is possible if the dimension of the embedding space is greater than five. In this context, the representation space $\mathbf{B}$ can be simplified with the 10-dimensional vector $\mathbf{b} \in \mathbb{R}^{10}$. Then we introduce the *Smooth Global Section Theorem* to show that the representation space is a smooth and continuous mapping to $SO(3)$, where the proof can be found in [Peretroukhin et al., 2020].

**Theorem 3** (Smooth Global Section). *Consider the surjective map $f: \mathbb{R}^{10} \to SO(3)$ such that $f(\mathbf{B})$ returns the rotation matrix $\mathbf{R}$ defined by the two antipodal unit quaternions $\pm \mathbf{q}_r$ by minimizing the QCQP problem. There exists a smooth and global mapping, or section, $h: SO(3) \to \mathbb{R}^{10}$ such that $f(h(\mathbf{R})) = \mathbf{R}$.*

## 5.2 GRADIENT COMPUTATION

The relationship of $\mathbf{q}_r$ and data matrix $\mathbf{B}$ is defined in terms of an objective and constraints in a mathematical optimization problem. Importantly, the derivative of $\mathbf{q}_r$ with respect to data matrix $\mathbf{B}$ follows the implicit differentiation [Gould et al., 2021].

Regarding the continuousness of the proposed intermediary representation space $\mathbf{F}$, we in this paper introduce a gradient computation method for the backpropagation procedure. Recall that the matrix $\mathbf{B} \in \mathbb{R}^{4 \times 4}$ is real symmetric, which can be simplified with a 10-dimensional vector

$\mathbf{b} = vec(\mathbf{B})$. Magnus [1985] demonstrated that $\mathbf{q}_r$ will be differentiable at $\mathbf{b}$ provided that the minimum eigenvalue $\lambda_1$ of $\mathbf{B}$ is simple[2]. Hence, the gradient is implemented using the *implicit function theorem*,

$$\frac{\partial \mathbf{q}_r}{\partial \mathbf{b}} = \mathbf{q}_r \otimes (\lambda_1 \mathbf{I} - \mathbf{B})^+, \tag{8}$$

where $\otimes$ denotes the Kronecker product, $(\cdot)^+$ refers to the Moore-Penrose pseudo-inverse.

## 5.3 UNCERTAINTY MEASUREMENT

The introduction of the uncertainty metric is of great significance to measure pose uncertainties. Intuitively, we consider the proposed unit dual quaternion distribution as the composition of the Bingham distribution $B(\mathbf{q}_r; \mathbf{B})$ and the Gaussian distribution $G(\mathbf{q}_d; -\mathbf{F}_3^{-1}\mathbf{F}_2^{\mathbf{T}}\mathbf{q}_r, -\frac{1}{2}\mathbf{F}_3^{-1})$.

For the rotation uncertainty, the Bingham belief is a proper choice [Peretroukhin et al., 2020]. According to the property of the Bingham distribution in Equation (5), the rotation uncertainty is written as

$$U_q(\mathbf{Z}) = \sum_i^4 z_i = z_1 + z_2 + z_3 - 3z_4, \tag{9}$$

where $z_i \leq 0, i = 1, 2, 3, 4$.

Likewise, we also use the Gaussian belief to measure the translation uncertainty. Here we decompose the covariance matrix $\mathbf{G} = -\frac{1}{2}\mathbf{F}_3^{-1}$ using the eigendecomposition method, and then the translation uncertainty can be written as

$$U_t(\mathbf{G}) = \sum_i^4 \lambda_i = \lambda_1 + \lambda_2 + \lambda_3 + \lambda_4, \tag{10}$$

where $\lambda_i, i = 1, 2, 3, 4$ denotes eigenvalues of $\mathbf{G}$.

---

[2]We find that the non-simple minimum eigenvalue occurs rarely in our work.

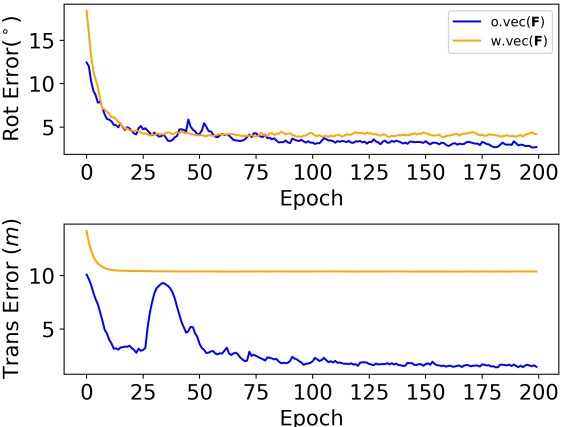

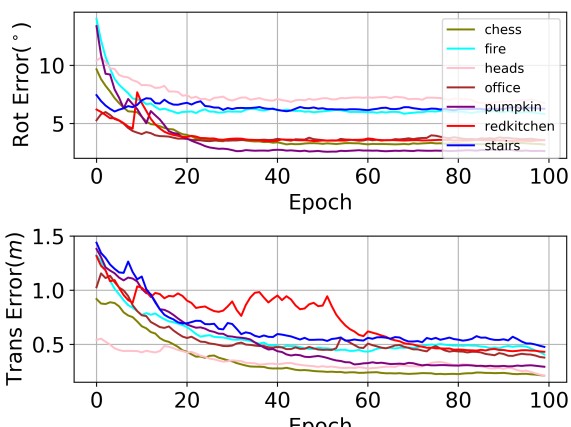

Figure 5: The difference of the use of the normalization of $vec(\mathbf{F})$ during the learning phase. We take the ShopFacade scene in the Cambridge Landmarks dataset for example. The orange line indicates the pose error curves without normalizing the $vec(\mathbf{F})$. While the blue line denotes the pose error curves after normalizing the $vec(\mathbf{F})$. o.$vec(\mathbf{F})$/w.$vec(\mathbf{F})$: without/with normalization of the $vec(\mathbf{F})$.

Figure 6: The rotation and translation error curves of each scene during test stages on the 7-Scenes dataset.

while $\mathbf{t}_g$ and $\mathbf{q}_g$ are the labeled translation and rotation. Moreover, we set the scale factor $\alpha = 100$ on the 7-Scenes dataset and $\alpha = 300$ on the Cambridge Landmarks dataset.

## 5.4 NETWORK STRUCTURE

The structure of the proposed deep probabilistic dual quaternion network is shown in Figure 3. We set the ResNet-50 as the backbone network where the parameters are initialized from pre-trained ImageNet weights. Then two fully connected layer are appended to the ResNet's activations to regress the parameter matrix $\mathbf{F}$. Since the parameter matrix $\mathbf{F}$ is real symmetric, we can encode it with a 36-dimensional feature vector. Subsequently, we decompose the $\mathbf{F}$ with two branches, namely rotation branch and translation branch. For the rotation branch, we estimate rotation quaternions by solving the QCQP problem shown in Figure 1. For the translation branch, we surprisingly find that directly using the estimated rotation quaternions is not sufficient to get a proper translation vector. In this case, a tiny trick is adopted that we normalize the output vector $vec(\mathbf{F})$ to generate a pseudo matrix $\mathbf{B}'$ which is shown in Figure 5. Then another rotation quaternion is estimated to compose the mean vector $\mathbf{m} = -\mathbf{F_3}^{-1}\mathbf{F_2^T}\mathbf{q_r}$. During the training stage, an implicit function theorem is applied to the backpropagation procedure. Finally, the translation vector $\mathbf{t}$ can be recovered from $\mathbf{m}$.

Additionally, we adopt the similar loss function presented in [Kendall et al., 2015] as our loss metric[3],

$$L = \|\mathbf{t}_g - \mathbf{t}_e\|_2 + \alpha\|\mathbf{q}_g - \mathbf{q}_e\|_2, \qquad (11)$$

where $\mathbf{t}_e$ and $\mathbf{q}_e$ denote the inferred translation and rotation,

---

[3]Unit quaternions $-\mathbf{q}$ and $\mathbf{q}$ represent the same rotation. Hence the difference between two unit quaternions is further detailed as $\|\mathbf{q}_g - \mathbf{q}_e\|_2 = \min(\|\mathbf{q}_g - \mathbf{q}_e\|_2, \|\mathbf{q}_g + \mathbf{q}_e\|_2)$.

## 6 EXPERIMENTS

In this section, we perform the absolute pose regression on the task of camera re-localization on two public datasets, namely Cambridge Landmarks [Kendall et al., 2015] and 7-Scenes datasets [Shotton et al., 2013], which consists of RGB frames with associated ground truth camera poses and provides training as well as test sequences. First, we give a comparison with state-of-the-art pose regression methods to demonstrate the inferred pose accuracy. Then we evaluate our deep probabilistic model on the noisy Cambridge Landmarks dataset to show its robustness in dealing with uncertainty and ambiguity.

### 6.1 TRAINING DETAILS

We run our experiments in the Pytorch framework [Paszke et al., 2019]. We use the Adam optimizer [Kingma and Ba, 2015] and begin with a learning rate of $10^{-4}$, and gradually decrease the learning rate exponentially with the multiplicative factor being 0.9. We use a batch size of 16 and train for 100 epochs for the 7-Scenes dataset and 200 epochs for the Cambridge Landmarks dataset. All input frames are resized to $224 \times 224$.

### 6.2 RESULTS

#### 6.2.1 Normal Scenes

**7-Scenes Dataset**. We test our model on all 7 scenes on the 7-Scenes dataset. Since the majority of the scenes do not

Table 1: Evaluation on the 7-Scenes dataset. The results are reported with the median translation error(m) and the median rotation error(°). The best results are in **bold**.

| Scene | Chess | Fire | Heads | Office | Pumpkin | RedKitchen | Stairs |
|---|---|---|---|---|---|---|---|
| PoseNet | 0.32m/8.12° | 0.47m/14.4° | 0.29m/12.0° | 0.48m/7.68° | 0.47m/8.42° | 0.59m/8.64° | 0.47m/13.8° |
| Dense PoseNet | 0.32m/6.60° | 0.47m/14.0° | 0.30m/12.2° | 0.48m/7.24° | 0.49m/8.12° | 0.58m/8.34° | 0.48m/13.1° |
| MapNet | **0.08m**/3.25° | 0.27m/11.69° | 0.18m/13.2° | **0.17m**/5.15° | 0.22m/4.02° | 0.23m/4.93° | 0.30m/12.08° |
| MapNet++ | 0.10m/3.17° | **0.20m**/9.04° | 0.13m/11.1° | 0.18m/5.38° | **0.19m**/3.92° | 0.20m/5.01° | 0.30m/13.4° |
| BPN | 0.37m/7.24° | 0.43m/13.7° | 0.31m/12.0° | 0.48m/8.04° | 0.61m/7.54° | 0.58m/7.54° | 0.48m/13.1° |
| VidLoc | 0.18m/- | 0.26m/- | 0.14m/- | 0.26m/- | 0.36m/- | 0.31m/- | **0.26m**/- |
| UBN | 0.10m/4.97° | 0.27m/12.87° | **0.12m**/14.05° | 0.20m/7.52° | 0.23m/7.11° | **0.19m**/8.25° | 0.28m/13.1° |
| MBN-MB | 0.10m/4.35° | 0.28m/11.86° | **0.12m**/12.76° | 0.19m/6.55° | 0.22m/6.9° | 0.21m/8.08° | 0.31m/9.98° |
| Ours | 0.20m/**2.9°** | 0.30m/**5.63°** | 0.19m/**6.53°** | 0.30m/**3.51°** | 0.28m/**2.6°** | 0.40m/**3.6°** | 0.42m/**6.23°** |

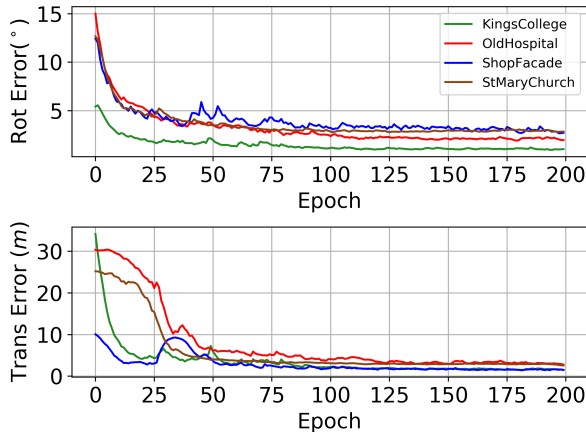

Figure 7: The rotation and translation error curves of each scene during test stages on the Cambridge Landmarks dataset.

show highly ambiguous environments, we regard them to be non-ambiguous. The final test curves of 7 scenes can be found in Figure 6. Clearly, the overall rotation error is less than 7° and the overall translation error is less than 0.5m.

In order to demonstrate the pose accuracy of our model, we make a comparison with other pose regression methods including PoseNet and its variant Dense PoseNet [Kendall et al., 2015], MapNet and its variant MapNet++ [Brahmbhatt et al., 2018], BPN [Kendall and Cipolla, 2016], VidLoc [Clark et al., 2017], UBN and MBN-MB [Deng et al., 2020]. The quantitative and qualitative results are listed in Table 1.

From Table 1, the evaluated results on the 7-Scenes dataset show that our method outperforms state-of-the-art methods on the rotation accuracy. But the translation accuracy performs a bit worse than baselines since the translation part is computed from the estimated rotation quaternion and the intermediary space $\mathbf{F}$, where the estimated rotation error can directly affect the translation accuracy and further amplify this error to the translation part. Despite this, our method still has a competitive advantage in terms of translation accuracy. A similar tendency also happens in the Cambridge

Landmarks dataset.

**Cambridge Landmarks Dataset**. To further demonstrate the pose accuracy on different scenes, we also implement our approach on the Cambridge Landmarks dataset. We select the Kings College, Hospital, ShopFacade and St.Mary Church as our evaluation scenes. Again, we plot the four different test curves in Figure 7. The final converge results show that the overall rotation error is less than 3° and overall translation error is no more than about 3m.

Next, we also list our final pose accuracy in Table 2 to make a comparison with state-of-the-art methods. Similar pose accuracy is reported that our method can achieve a more accurate pose in the Cambridge Landmarks dataset especially for the rotation part.

### 6.2.2 Noisy Scenes

To further demonstrate the performance of our method in dealing with uncertainty and ambiguity, we conduct our model on four noisy scenarios, namely the Kings College, Hospital, ShopFacade, and St.Mary Church, where these scenes are processed by manually adding the Gaussian blur kernel, randomly changing the brightness, contrast, saturation of all frames, and both to simulate different ambiguous environments.

Without retraining the proposed deep probabilistic dual quaternion network, we directly feed the processed frames into the trained model to predict the camera pose. The quantitative results are shown in Table 3. The results show that the pose errors have some minor changes in noisy scenes including the Kings College, Hospital, ShopFacade and St.Mary Church.

Then we measure uncertainties of our model under normal and noisy environments, which is shown in Figure 8. Intuitively, the pose errors in the blur environment denoted by purple points have a similar distribution compared to that in the original environment denoted by the red points under the both rotation uncertainty and translation uncertainty measurement. Nevertheless, the pose errors in the

Table 2: Evaluation on the Cambridge Landmarks dataset. The results are reported with the median translation error(m) and the median rotation error(°). The best results are in **bold**.

| Scene | Kings College | Hospital | ShopFacade | St.Mary Church |
|---|---|---|---|---|
| PoseNet | 1.92m/5.40° | 2.31m/5.38° | 1.46m/8.08° | 2.65m/8.48° |
| Dense PoseNet | 1.66m/4.86° | 2.62m/4.90° | 1.41m/7.18° | 2.45m/7.96° |
| MapNet | 1.07m/1.89° | 1.94m/3.91° | 1.49m/4.22° | 2.0m/4.53° |
| BPN | 1.74m/4.06° | 2.57m/5.12° | 1.25m/7.54° | 2.11m/8.38° |
| UBN | 0.88m/1.77° | **1.93m**/3.71° | **0.8m**/4.74° | 1.84m/6.19° |
| MBN-MB | **0.83m**/2.08° | 2.16m/3.64° | 0.92m/4.93° | **1.37m**/6.03° |
| Ours | 1.20m/**0.84°** | 2.46m/**1.72°** | 1.10m/**2.51°** | 2.40m/**2.63°** |

Table 3: The quantitative results of the proposed deep probabilistic dual quaternion network in the noisy scenes on the Cambridge Landmarks dataset.

| Scene | Kings College | Hospital | ShopFacade | St.Mary Church |
|---|---|---|---|---|
| Normal | 1.20m/0.84° | 2.46m/1.72° | 1.10m/2.51° | 2.4m/2.63° |
| Blur | 1.42m/0.92° | 2.59m/1.74° | 1.12m/2.53° | 2.60m/2.65° |
| Brightness | 1.55m/1.00° | 3.07m/2.14° | 1.30m/2.61° | 3.00m/2.60° |
| Blur & Brightness | 1.76m/1.22° | 3.29m/2.31° | 1.38m/2.86° | 3.13m/2.69° |

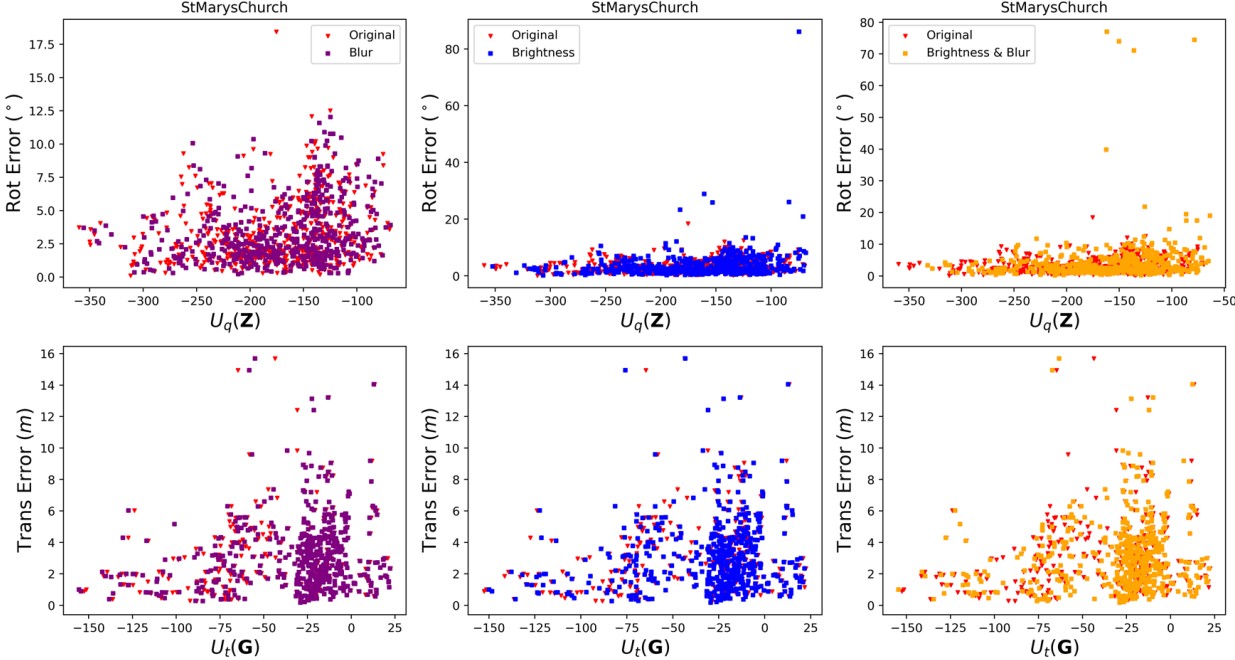

Figure 8: Uncertainty evaluation on the Cambridge Landmarks dataset. The left column shows the pose errors under the pose uncertainty metric in the blur environment, where the radius of the Gaussian blur kernel is $3.8$. The middle column shows the pose errors under the pose uncertainty metric in the random brightness change environment, where the maximum brightness factor is $0.6$, the maximum contrast factor is $0.6$ and the maximum saturation factor is $0.5$. The right column indicates pose errors under the both blur and brightness change environment. Note: we only plot the St.Mary Church scene, full information can be found in the Supplementary Material.

brightness change environment denoted by blue points have some minor differences especially for the rotation uncertainty. Likewise, the pose errors in the blur and brightness change environment denoted by the orange points have a similar tendency, but we believe that it is reasonable in noisy environments. More importantly, there are only a few points out of the original distribution in aforementioned two scenes which have limited effects to the overall pose accuracy. Furthermore, the pose accuracy of our method in noisy environments still outperforms the state-of-the-arts that in normal environments especially for the rotation accuracy. As a result, the experiment results in noisy environments suggest that our model is robust to deal with uncertainty and ambiguity.

# 7 CONCLUSION

We design a deep probabilistic dual quaternion network that addresses the absolute pose regression problem on $SE(3)$. Unlike existing work, we take pose uncertainties into consideration by introducing an antipodally symmetric distribution over the unit dual quaternion on $SE(3)$. To address the mapping problem from the Euclidean space $\mathbb{R}^n$ to $SE(3)$ manifold, we present an intermediary differential representation space $\mathbf{F}$ as the output of our model to indirectly regress poses. Additionally, we introduce a backpropagation method for batch optimization. Experiment results on the camera relocalization task on the 7-Scenes dataset and the Cambridge Landmarks dataset show that our method outperforms state-of-the-art methods on the pose accuracy. Moreover, extensive experiments on the noisy scenes on the Cambridge Landmarks dataset show that our method has the ability to deal with uncertainty and ambiguity.

In the future, we will explore the absolute pose regression problem leveraging our representation with a negative log-likelihood loss function to improve the reliability and robustness of our model in real-world applications.

## Acknowledgements

This research is financially supported by the National Natural Science Foundation of China (No. 62072231), Fundamental Research Funds for the Central Universities (No. 14380079), and the Collaborative Innovation Center of Novel Software Technology and Industrialization. Jia Liu (jialiu@nju.edu.cn) and Lijun Chen (chenlj@nju.edu.cn) are the corresponding authors.

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
