# OpenReview forum: "PDQ-Net: Deep Probabilistic Dual Quaternion Network for Absolute Pose Regression on $SE(3)$"
_auai.org/UAI/2022/Conference — UAI 2022 Poster_

### Official Review · Reviewer_688M · 2022-04-09

**Q2(1) Originality/Novelty:** 3
**Q2(2) Significance/Impact:** 2
**Q2(3) Correctness/Technical Quality:** 3
**Q2(6) Clarity Of Writing:** 4
**Q6 Overall Score:** 7
**Q8 Confidence In Your Score:** 2

**Q1 Summary And Contributions:**

The authors propose a novel probabilistic dual quaternion network that indirectly regresses the pose of a given scene. To this end, it models uncertainty via an antipodally symmetric probability distribution over the unit dual quaternions on the SE(3) manifold. In their experiments, the authors could show advantageous generalization performance with respect to rotation errors compared to recent pose estimation baselines.

**Q2 Assessment Of The Paper:**

More detailed information regarding each of these aspects is given below:

**Q2(4) Quality Of Experiments (Optional):**

3: Good: The experimental evaluation is adequate, and the results convincingly support the main claims.

**Q2(5) Reproducibility:**

2: Fair: Key resources (e.g., proofs, code, data) are unavailable but key details (e.g., proof sketches, experimental setup) are sufficiently well-described for an expert to confidently reproduce the main results.

**Q3 Main Strengths:**

- Elegant embedding of uncertainty quantification in the applied dual quaternion framework
- Despite its theoretical complexity, the model is mapped to an end-to-end learning framework
- Substantial improvements in the rotation error, somewhat competitive performance with respect to the translation error at the same time

**Q4 Main Weakness:**

- SOTA baselines missing
- No proper hyperparameter optimization for fair comparisons
- Unclear statistical significance of the results

**Q5 Detailed Comments To The Authors:**

Major:
- Turkoglu et al., 2021 (“Visual Camera Re-Localization Using Graph Neural Networks and Relative Pose Supervision”) gives a good overview over recent pose estimation baselines and their performances. As can be seen, Table 1 in their paper lists baselines that are better than the presented results in the reviewed paper (albeit only marginally). The SOTA baselines should be also considered within the empirical evaluation.
- For a fair comparison, all assessed combinations should involve a proper hyperparameter optimization for a fair comparison. Right now, it is unclear whether the chosen parameters are cherry-picked and in favor with the presented approach.
- For statistical significance, the experiments should be conducted several times and the statistical significance of the results should be determined.

Minor:
- I think it would be good to elaborate a bit more on the absolute pose estimation problem as the UAI audience might not be familiar with it.
- Some spelling mistakes (e.g., “Repersentation” in Figure 4)
- Please mark best results in the tables
- Table 3: How do baselines perform on noisy scenes?
- What could be potential reasons for the worse translation error performance of the proposed model?


**Q7 Justification For Your Score:**

The approach demonstrates an elegant uncertainty quantification with a proper theoretical modelling and the empirical results are convincing enough (despite the described shortcomings).

**Q9 Complying With Reviewing Instructions:**

1: Yes.

---

### Official Review · Reviewer_7TPK · 2022-04-12

**Q2(1) Originality/Novelty:** 2
**Q2(2) Significance/Impact:** 2
**Q2(3) Correctness/Technical Quality:** 3
**Q2(6) Clarity Of Writing:** 2
**Q6 Overall Score:** 5
**Q8 Confidence In Your Score:** 3

**Q1 Summary And Contributions:**

The paper proposes a new deep learning method for pose regression in SE(3) (accounting for translation + rotation of the view point).
They develop a bi-polar distribution on the unit dual quaternion space accounting for uncertainties in the pose estimation, elaborate a differentiable parameterization of this representation, and test it on two classical benchmarks of camera relocalization, where the method performs well.

**Q2 Assessment Of The Paper:**

More detailed information regarding each of these aspects is given below:

**Q2(4) Quality Of Experiments (Optional):**

3: Good: The experimental evaluation is adequate, and the results convincingly support the main claims.

**Q2(5) Reproducibility:**

3: Good: Key resources (e.g., proofs, code, data) are available and key details (e.g., proofs, experimental setup) are sufficiently well-described for competent researchers to confidently reproduce the main results.

**Q3 Main Strengths:**

- extension of the probability distribution on the unit dual quaternion hypersphere from SE(2) (planar motion) to SE(3)
- differentiable framework (mostly Section 5)
- extensive experiments with SOTA competitors

**Q4 Main Weakness:**

- the extension from SE(2) to SE(3) seems quite immediate, and probably already explored in related works (see detailed comments for references)
- the description of the differentiable framework and use of implicit function theorem lacks rigor
- under performances on the translation estimation of the method

**Q5 Detailed Comments To The Authors:**

I believe that the paper is a sensible step toward the use of dual quaternion representation in pose estimation with deep learning.
Yet, connections with missing related works should be deepen.

Minor comments :
- the use of exponential map in Eq. (1) (and the Riemannian structure of the considered manifold) should be detailed.
- The formulation of the QCQP in Eq. (7) seem strange to me. The minimization is already conducted on the Unit Hypersphere S^3, which is redundant with the constraint. Why not solving this problem with Riemannian gradient descent [1] ?
- Section 5.2 lacks rigor in its description. The use (and the conditions for applicability) of the implicit function theorem should be better described.
- Regarding the loss, why not choosing a geodesic distance between rotations instead of a Euclidean distance ?
- Scores obtained on the translation part are to a certain extent below than SOTA, but better in terms of orientation. Is it linked to the choice of the \alpha coefficient in the loss ? Maybe this point could have been discussed in the paper.

Missing related works :
Zhang, L.; Shang, H.; Lin, Y. A Novel Distribution for Representation of 6D Pose Uncertainty. Micromachines 2022, 13, 126. https://doi.org/10.3390/mi13010126

I. Gilitschenski, G. Kurz, S. J. Julier and U. D. Hanebeck, "A new probability distribution for simultaneous representation of uncertain position and orientation," 17th International Conference on Information Fusion (FUSION), 2014, pp. 1-7.


[1] Optimization Algorithms on Matrix Manifolds P.-A. Absil, R. Mahony, R. Sepulchre Princeton University Press, 2008


[After rebuttal] I thank the authors for taking my remarks into account in their rebuttal.

**Q7 Justification For Your Score:**

I believe that the paper is a sensible step toward the use of dual quaternion representation in pose estimation with deep learning.
I noted some missing related works that should be included in the paper. My rating leans toward a borderline accept since overall:
 - the contribution is somehow limited or straightforward
 - in terms of writing, the paper is ok to read but somehow lacks of rigor in some parts of the exposition



**Q9 Complying With Reviewing Instructions:**

1: Yes.

---

### Official Review · Reviewer_jifS · 2022-04-15

**Q2(1) Originality/Novelty:** 3
**Q2(2) Significance/Impact:** 3
**Q2(3) Correctness/Technical Quality:** 3
**Q2(6) Clarity Of Writing:** 3
**Q6 Overall Score:** 6
**Q8 Confidence In Your Score:** 2

**Q1 Summary And Contributions:**

The paper develops a exponential quaternion distribution which is expanded as combination of Bingham and Gaussian like matrices, with which one is able to solve for rotation and translation respectively through Quadratic programming and Gaussian Processes respectively; with a backpropagation setup derived from the inverse function theorem. In addition to comparisons, the paper also examines robustness to uncertainty (blur, lighting).

**Q2 Assessment Of The Paper:**

More detailed information regarding each of these aspects is given below:

**Q2(4) Quality Of Experiments (Optional):**

3: Good: The experimental evaluation is adequate, and the results convincingly support the main claims.

**Q2(5) Reproducibility:**

2: Fair: Key resources (e.g., proofs, code, data) are unavailable but key details (e.g., proof sketches, experimental setup) are sufficiently well-described for an expert to confidently reproduce the main results.

**Q3 Main Strengths:**

Uncertainty measurements are an important aspect of pose estimation and this paper provides a way to do this, considering related works such as the paper proposing use of Bingham loss. I felt that the derivation and use of components with quadratic optimization and the inverse function backpropagation rule quite elegant. Evaluations compare favourably with other competitive methods. The experiment on robustness to uncertainty is also quite convincing.

**Q4 Main Weakness:**

While the method is explained quite clearly, the connections motivations for using some of the components the way they have are not clear upon immediate perusal - this is purely subjective. For instance, how does the method improve upon or compare with the Bingham distribution formulation in Gilitschenski et al [1]?

In the evaluations, robustness experiments are shown with the authors concluding that there is no significant difference between the perturbed setup vs the original one. How do other methods compare in this regard?

[1] https://openreview.net/pdf?id=ryloogSKDS

**Q5 Detailed Comments To The Authors:**

I think the presentation is good in an overall sense. However, key intuitions could be explained better, such as the dual quaternion formulation, the idea of antipodal symmetry, etc.

**Q7 Justification For Your Score:**

The work builds upon existing works containing the Bingham distribution to develop models for uncertainty in pose and translation in a principled way and goes on to show the effectiveness of the model through sensible experiments. The focus on robustness to uncertainty seems to be quite significant.

**Q9 Complying With Reviewing Instructions:**

1: Yes.

---

### Official Review · Reviewer_FGTx · 2022-04-16

**Q2(1) Originality/Novelty:** 3
**Q2(2) Significance/Impact:** 3
**Q2(3) Correctness/Technical Quality:** 3
**Q2(6) Clarity Of Writing:** 3
**Q6 Overall Score:** 7
**Q8 Confidence In Your Score:** 3

**Q1 Summary And Contributions:**

This paper presents a deep probabilistic model for absolute pose regression on SE(3). The key contributions include 1) dual quaternion parameterization; 2) a unit dual quaternion distribution on SE(3); 3) a deep neural network based parameterization of the proposed distribution. Experiments on two real-world datasets are performed to verify the effectiveness of the proposed model.


**Q2 Assessment Of The Paper:**

More detailed information regarding each of these aspects is given below:

**Q2(4) Quality Of Experiments (Optional):**

3: Good: The experimental evaluation is adequate, and the results convincingly support the main claims.

**Q2(5) Reproducibility:**

3: Good: Key resources (e.g., proofs, code, data) are available and key details (e.g., proofs, experimental setup) are sufficiently well-described for competent researchers to confidently reproduce the main results.

**Q3 Main Strengths:**

1. The paper is clearly written. In particular, the multiple diagrams are very helpful in understanding the key ideas.

2. The technical contribution is significant. The proposed exponential distribution solves the previous challenge of defining a distribution for representing the position and orientation on the SE(3) manifold. Moreover, the authors propose the intermediary differential representation space which allows efficient back-propagation via implicit function theorem.

3. The experiments on two real-world datasets show that the proposed method is superior to state-of-the-art methods in terms of orientation estimation.


**Q4 Main Weakness:**

1. The experimental results on translation error are worse than methods like MapNet. Could you provide more discussion about why this is the case? Especially, it would be great to at least discuss why the proposed method helps more in estimating the orientation compared to the translation.

2. In Section 6.2.2., how do you exactly achieve the random change of the brightness, contrast, and saturation of all frames? It seems not surprising that the proposed method could work well under these synthetic noisy scenes if the manually added randomness is more or less Gaussian type. It would be interesting to explore more realistic noisy scenarios.


**Q5 Detailed Comments To The Authors:**

Please see my comments above.

**Q7 Justification For Your Score:**

Based on the comments above, I tend to accept the paper.

**Q9 Complying With Reviewing Instructions:**

1: Yes.

---

### Decision · Program_Chairs · 2022-05-15

**Decision:**

Accept (Poster)

**Comment:**

Meta Review: This paper introduces a deep learning method for scene pose estimation.  It received four mostly positive reviews, including  two accepts, one weak accept, and one borderline accept.  Reviewers acknowledge the paper’s technical contributions, including it’s elegant formulation,  uncertainty quantification,  end-to-end learning,  and the method’s competitive performance in rotation estimation.  There are some concerns, including inferior performance in translation errors  and missing  comparison with some SOTA methods.  During rebuttal, the authors adequately address most of the concerns.